# Simple Randomized Rounding for Max-Min Eigenvalue Augmentation

**Jourdain Lamperski** [1]  **Haeseong Yang** [1]  **Oleg A. Prokopyev** [1,2]

## Abstract

We consider the *max-min eigenvalue augmentation* problem: given $n \times n$ symmetric positive semidefinite matrices $M, A_1, \ldots, A_m$ and a positive integer $k < m$, the goal is to choose a subset $I \subset \{1, \ldots, m\}$ of cardinality at most $k$ that maximizes the minimum eigenvalue of the matrix $M + \sum_{i \in I} A_i$. The problem captures both the *Bayesian E-optimal design* and *maximum algebraic connectivity augmentation* problems. In contrast to the existing work, we do not assume that the *augmentation matrices* are rank-one matrices, and we focus on the setting in which $k < n$. We show that a *simple* randomized rounding method provides a constant-factor approximation if the *optimal increase* is sufficiently large, specifically, if $\mathrm{OPT} - \lambda_{\min}(M) = \Omega(R \ln k)$, where OPT is the optimal value, and $R$ is the maximum trace of an augmentation matrix. To establish the guarantee, we derive a matrix concentration inequality that is of independent interest. The inequality can be interpreted as an *intrinsic dimension* analog of the matrix Chernoff inequality for the minimum eigenvalue of a sum of independent random positive semidefinite matrices; such an inequality has already been established for the maximum eigenvalue, but not for the minimum eigenvalue.

## 1. Introduction

Let $S^n_+$ denote the set of symmetric positive semidefinite (PSD) matrices. Given $M \in S^n_+$, *augmentation matrices* $A_1, \ldots, A_m \in S^n_+$, and a positive integer $k < m$, we con-

[1]Department of Industrial Engineering, University of Pittsburgh, 1025 Benedum Hall, Pittsburgh, PA 15261, USA [2]Department of Business Administration, University of Zurich, Plattenstrasse 14, CH 8032 Zurich. Correspondence to: Jourdain Lamperski <lamperski@pitt.edu>.

*Proceedings of the 42nd International Conference on Machine Learning*, Vancouver, Canada. PMLR 267, 2025. Copyright 2025 by the author(s).

sider the *max-min eigenvalue augmentation* problem:

$$\max_{z \in \{0,1\}^m} \left\{ \lambda_{\min}\left( M + \sum_{i=1}^m z_i A_i \right) \ : \ \|z\|_0 \le k \right\}, \quad (1)$$

where $\lambda_{\min}\left( M + \sum_{i=1}^m z_i A_i \right)$ denotes the minimum eigenvalue of the matrix $M + \sum_{i=1}^m z_i A_i$, and $\|z\|_0$ is equal to the number of non-zero entries in $z$. Problem (1) captures the *Bayesian E-optimal design* problem and the *maximum algebraic connectivity augmentation* problem, which we outline in more detail below.

**Bayesian E-optimal design.** When running experiments to collect data to build a statistical model, it is naturally of interest to run fewer experiments to obtain the same level of "model quality." The goal in *optimal linear experimental design* is to determine which experiments to run in order to minimize the "variance" of a linear regression model, subject to some cardinality constraint on the number of the experiments (Fedorov, 2013; Goos & Jones, 2011; Pukelsheim, 2006). (Note that linear regression models are unbiased under mild conditions (Hastie et al., 2009).)

More formally, consider *design points* $x_1, \ldots, x_m \in \mathbb{R}^n$ that each correspond to a potential experiment. If we run the experiment associated with design point $x_i$, then we observe an *outcome* $y_i \in \mathbb{R}$. Suppose that we can run at most $k$ experiments. Running the experiments associated with design points $x_i$, $i \in I \subset [m] := \{1, \ldots, m\}$ (where $|I| \le k$) provides us with data $\{(x_i, y_i)\}_{i \in I}$, which then can be used to construct a linear regression model. The goal in optimal linear experimental design is to select a subset $I$ that optimizes a function $f$ (referred to as an *optimality criterion*) of the *information matrix* $\sum_{i \in I} x_i x_i^\top$. There are a number of optimality criteria; each captures a different notion of variance.

The goal in E-optimal design is to maximize the *E-optimality criterion*, which is given by the minimum eigenvalue $\lambda_{\min}(\sum_{i \in I} x_i x_i^\top)$ of the information matrix. E-optimal design is a special case of (1) with $M = 0$ and augmentation matrices $A_i = x_i x_i^\top$, $i \in [m]$. In Bayesian E-optimal design, we are also given a *prior matrix*, which translates to an instance of (1) with $M \neq 0$ and the augmentation matrices given by $A_i = x_i x_i^\top$, $i \in [m]$. This setup assumes that the experiments can be run individually and hence, the augmentation matrices are each of rank one. If the

experiments can only be run in certain batches, then, as in the previous application setting, we must consider augmentation matrices of general rank (Revilla Sancho; Derezinski et al., 2020; Tantipongpipat, 2020; Che et al., 2024).

**Maximum algebraic connectivity augmentation.** In the maximum algebraic connectivity augmentation problem, the goal is to add at most $k$ edges to a given graph $G = (V, E)$ in order to maximize the graph's *algebraic connectivity* (Ghosh & Boyd, 2006; Kim, 2009; Kolla et al., 2010; Mosk-Aoyama, 2008; Somisetty et al., 2024; Wei et al., 2014). The algebraic connectivity of a graph is the second smallest eigenvalue $\lambda_2(L)$ of its Laplacian matrix $L = \sum_{e \in E} a_e a_e^\top$, where $a_e \in \mathbb{R}^n$ for each edge $e = ij \in E$ is defined by $a_e = e_i - e_j$, where $e_i$ and $e_j$ are the $i$-th and $j$-th standard $n$-dimensional unit vectors, respectively (De Abreu, 2007; Fiedler, 1973).

Algebraic connectivity provides a measure of a graph's connectivity. In particular, $\lambda_2(L) > 0$ if and only if $G$ is connected. Furthermore, larger values of $\lambda_2(G)$ indicate that $G$ is, in a sense, more connected. Indeed, algebraic connectivity appears as a factor in the convergence rates of various dynamic processes (e.g., Markov chains) on graphs (Ogiwara et al., 2015). Intuitively, one would expect a dynamic process to converge faster (e.g., a Markov chain to mix quicker) if the graph is, to some extent, more connected. Algebraic connectivity provides us with exactly this notion of connectivity.

The maximum algebraic connectivity augmentation problem finds applications in settings in which it is of interest to add edges to a graph in order to make the graph more robust with respect to potential edge disruptions. Consider, for example, adding flight routes to an air transportation network in order to improve its resiliency to bad weather and airport closures (Wei et al., 2014).

More formally, it can be shown that $\lambda_2(L) = \lambda_{\min}(L + \mathbf{1}\mathbf{1}^\top)$, where $\mathbf{1}$ is the $n$-dimensional vector of all ones; we provide a short proof of this fact in Appendix B. Accordingly, we can formulate the maximum algebraic connectivity augmentation problem as a special case of (1) with $M = L + \mathbf{1}\mathbf{1}^\top$ and the augmentation matrices given by $A_e = a_e a_e^\top$, $e \notin E$. The outlined setup assumes that the edges can be individually added, which gives rise to rank-one augmentation matrices. If the edges must be added in groups (e.g., as particular subgraphs), then one must consider augmentation matrices of general rank.

**Simple randomized rounding.** The max-min eigenvalue augmentation problem (1) is $NP$-hard because the maximum algebraic connectivity augmentation problem is $NP$-hard (Mosk-Aoyama, 2008). The E-optimal design problem is similarly $NP$-hard (Civril & Magdon-Ismail, 2009).

In this work, we study a simple randomized rounding

method for max-min eigenvalue augmentation. More precisely, consider the following semidefinite programming relaxation of (1):

$$\max_{z \in [0,1]^m, \eta \in \mathbb{R}} \left\{ \eta : M + \sum_{i=1}^m z_i A_i \succeq \eta I \text{ and } \sum_{i=1}^m z_i \leq k \right\},$$
(2)

where $I$ is the $n \times n$ identity matrix. Let $(z_{\mathrm{sdp}}, \eta_{\mathrm{sdp}}) \in [0,1]^m \times \mathbb{R}$ denote an optimal solution to the relaxation (2). We study the randomized method that, for each $i \in [m]$, rounds the $i$-th entry of $z_{\mathrm{sdp}}$ to be equal to 1 with probability $[z_{\mathrm{sdp}}]_i$ and 0, otherwise. In other words, we study the random vector $z_{\mathrm{round}} \in \{0,1\}^m$ defined by

$$\mathbb{P}([z_{\mathrm{round}}]_i = 1) = [z_{\mathrm{sdp}}]_i, \quad \forall i \in [m].$$

Note that $z_{\mathrm{round}}$ might not be feasible for (1) because $\sum_{i=1}^m [z_{\mathrm{round}}]_i > k$ with positive probability; recall that $k < m$.

The maximum algebraic connectivity augmentation (Kolla et al., 2010; Wei et al., 2014) and E-optimal design (Allen-Zhu et al., 2021; Avron & Boutsidis, 2013; Lau & Zhou, 2022) studies also investigate approximately solving the max-min eigenvalue augmentation problem (1). Our study is different from these works in the following two key ways. *First*, unlike these studies, we do not assume that the augmentation matrices $A_1, \ldots, A_m$ are rank-one matrices, which allows us to capture more practical considerations (e.g., situations in which edges cannot be individually added to a graph, or experiments must be run in certain batches). *Second*, we focus on the regime in which $k < n$. The (non-Bayesian) E-optimal design studies assume that $k \geq n$; otherwise, the E-optimal design problem is not interesting because $\lambda_{\min}(\sum_{i \in I} x_i x_i^\top) = 0$ for all $I \subset [m]$ such that $|I| < n$. To the best of our knowledge, the Bayesian E-optimal design problem remains unexplored in the $k < n$ setting. In contrast, the existing maximum algebraic connectivity augmentation work considers the case when $k < n$. However, in the algebraic connectivity setting, the augmentation matrices are of a particular form; recall our earlier discussion on their structure. Thus, the maximum algebraic connectivity augmentation problem does not capture the full generality of the max-min eigenvalue augmentation problem (1), even under the assumption that the augmentation matrices are rank-one matrices.

Finally, as we discuss in more detail in Subsection 1.1, our study boils down to understanding the extent to which we can (in some very particular sense) approximate a PSD matrix with a sum of random PSD matrices in terms of the *intrinsic dimension* of the matrix being approximated. The results that we develop in this direction are more generally applicable and are of independent interest.

## 1.1. Summary of approach, contributions, and organization

**Approach.** Let OPT denote the optimal objective function value of the max-min eigenvalue augmentation problem (1). Define

$$\text{INC} := \lambda_{\min}\Big(M + \sum_{i=1}^{m}[z_{\text{sdp}}]_i A_i\Big) - \lambda_{\min}(M)$$

to be the increase in the minimum eigenvalue provided by $z_{\text{sdp}}$. Note that INC is an upper bound on the *optimal increase*. That is, $\text{INC} \geq \text{OPT} - \lambda_{\min}(M)$. We show that if INC (or more weakly, the optimal increase) is sufficiently large, then simple randomized rounding provides a constant-factor approximation.

We exploit the following observation. If INC is larger, then the quadratic form $x^\top (\sum_{i=1}^{m}[z_{\text{sdp}}]_i A_i)x$ can only be on the order of INC in a smaller (actually no larger, but we are speaking informally) number of *directions* $x \in S^{n-1} = \{x \in \mathbb{R}^n : \|x\| = 1\}$, where $\|x\|$ is the Euclidean norm of $x$. (We refer to points on the unit sphere as directions.) More precisely, for $0 < \gamma < 1$, the set

$$S\Big(\sum_{i=1}^{m}[z_{\text{sdp}}]_i A_i, \gamma\text{INC}\Big)$$
$$:= \Big\{x \in S^{n-1} : x^\top\Big(\sum_{i=1}^{m}[z_{\text{sdp}}]_i A_i\Big)x \geq \gamma\text{INC}\Big\} \quad (3)$$

of directions is smaller when INC is larger; see Figure 1 for an illustration.

Hence, when INC is larger, it is sufficient to ensure that $\sum_{i=1}^{m}[z_{\text{round}}]_i A_i$ is a good approximation of $\sum_{i=1}^{m}[z_{\text{sdp}}]_i A_i$ with respect to a smaller number of directions. (For $x \notin S(\sum_{i=1}^{m}[z_{\text{sdp}}]_i A_i, \gamma\text{INC})$, the quadratic form $x^\top M x$ must already be large, so we do not need to account for these directions.) More formally, for some $\epsilon \in (0, 1)$, it is sufficient to ensure that

$$x^\top\Big(\sum_{i=1}^{m}[z_{\text{round}}]_i A_i\Big)x \geq (1-\epsilon)\gamma\text{INC},$$
$$\forall x \in S\Big(\sum_{i=1}^{m}[z_{\text{sdp}}]_i A_i, \gamma\text{INC}\Big) \quad (4)$$

with some amount of probability. One would imagine that it is "easier" to ensure (4) when $S(\sum_{i=1}^{m}[z_{\text{sdp}}]_i A_i, \gamma\text{INC})$ is smaller, i.e., when INC is larger.

We show that we can ensure (4) with a higher amount of probability when the set $S(\sum_{i=1}^{m}[z_{\text{sdp}}]_i A_i, \gamma\text{INC})$ is smaller. Towards this end, we measure the size of the set

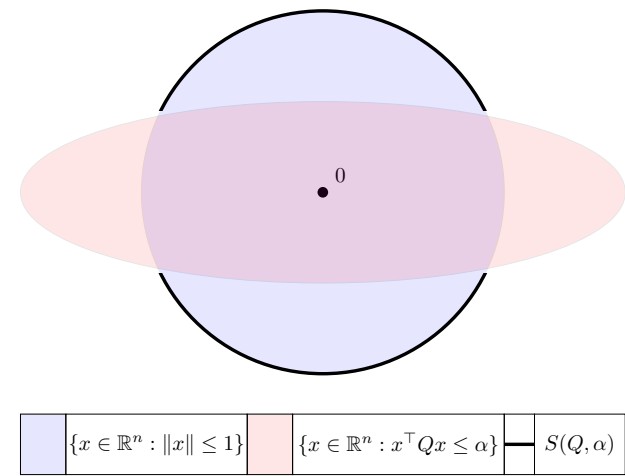



| $\{x \in \mathbb{R}^n : \|x\| \leq 1\}$ | $\{x \in \mathbb{R}^n : x^\top Q x \leq \alpha\}$ | $S(Q, \alpha)$ |



*Figure 1.* Illustration of set $S(Q, \alpha)$, where $Q$ is a positive definite matrix and $\alpha \geq 0$.

with the quantity

$$\text{intdim}\Big(\sum_{i=1}^{m}[z_{\text{sdp}}]_i A_i, \gamma\text{INC}\Big) := \frac{\text{tr}\Big(\sum_{i=1}^{m}[z_{\text{sdp}}]_i A_i\Big)}{\gamma\text{INC}}, \quad (5)$$

which we refer to as the *intrinsic dimension of $[z_{\text{sdp}}]_i A_i$ relative to $\gamma\text{INC}$*, and we lower-bound the probability of (4) in terms of $\text{intdim}(\sum_{i=1}^{m}[z_{\text{sdp}}]_i A_i, \gamma\text{INC})$.

More generally, we consider a generic sum $X = \sum_{i=1}^{m} X_i$ of independent random PSD matrices $X_1, \ldots, X_m \in S_+^n$. We develop an upper bound for

$$\mathbb{P}(\exists x \in S(\mathbb{E}[X], \alpha) : x^\top X x \leq (1-\epsilon)\alpha) \quad (6)$$

in terms of $\text{intdim}(\mathbb{E}[X], \alpha)$, where $\alpha > 0$. Suppose that $\alpha = \lambda_{\min}(\mathbb{E}[X])$ (in which case $\alpha > 0$ if $\mathbb{E}[X]$ is positive definite). Then, $S(\mathbb{E}[X], \alpha) = S^{n-1}$, and it follows that the probability given in (6) equals

$$\mathbb{P}(\exists x \in S^{n-1} : x^\top X x \leq (1-\epsilon)\lambda_{\min}(\mathbb{E}[X]))$$
$$= \mathbb{P}(\lambda_{\min}(X) \leq (1-\epsilon)\lambda_{\min}(\mathbb{E}[X])).$$

Accordingly, we, in some sense, develop an "intrinsic dimension" lower-tail bound for the minimum eigenvalue $\lambda_{\min}(X) = \min_{x \in S^{n-1}} x^\top X x$. Although, it is more precise to state that we develop an intrinsic dimension lower-tail bound for the quantity $\min_{x \in S(\mathbb{E}[X],\alpha)} x^\top X x$. Furthermore, we note that the tail bound that we develop is only interesting when $\alpha > \lambda_{\min}(\mathbb{E}[X])$.

**Contributions and organization.** First, we provide some background on matrix Chernoff inequalities for $\lambda_{\min}(X)$ and $\lambda_{\max}(X)$ in Section 2. In particular, we point out that an intrinsic dimension matrix Chernoff inequality has been established for $\lambda_{\max}(X)$, but not for $\lambda_{\min}(X)$. As stated

above, our work develops an intrinsic dimension analog of the matrix Chernoff inequality for $\lambda_{\min}(X)$. In Section 2 we also provide a reformulation of the existing matrix Chernoff inequality for $\lambda_{\min}(X)$ that plays a key role in our analysis; see Theorem 2.5 for a statement of the reformulation.

We derive the intrinsic dimension concentration inequality for $\lambda_{\min}(X)$ in Section 3; see Theorem 3.1 for a statement of the inequality. The derivation boils down to a careful application of the existing matrix Chernoff inequalities (including the reformulated inequality provided in Theorem 2.5) with respect to subsets of directions that lie in certain subspaces.

In Section 4 we use the intrinsic dimension concentration inequality stated in Theorem 3.1 to show that simple randomized rounding provides a constant-factor approximation if $\mathrm{INC} = \Omega(R \ln k)$, where $R := \max_{i \in [m]} \mathrm{tr}(A_i)$; see Theorem 4.1. Our proof of this result follows the argument provided above in the summary of our approach. If we instead applied the existing matrix Chernoff inequality for $\lambda_{\min}(X)$, then we would obtain the condition $\mathrm{INC} = \Omega(W \ln n)$, where $W := \max_{i \in [m]} \lambda_{\max}(A_i)$. This condition, however, does not shed light on the setting of interest in which $k < n$.

## 2. Background and preliminaries

First, in Subsection 2.1, we provide background on matrix Chernoff inequalities for the minimum and maximum eigenvalue of a sum of random PSD matrices. Then, in Subsection 2.2, we present a reformulation of the inequality for the minimum eigenvalue.

### 2.1. Matrix Chernoff inequalities

All inequalities that we present in this subsection are taken from Tropp (Tropp et al., 2015). Consider the following matrix Chernoff inequality for the maximum eigenvalue of a sum of independent random PSD matrices that are surely bounded (in terms of their maximum eigenvalue).

**Theorem 2.1** ((Tropp et al., 2015), Theorem 5.1.1). *Let $X_1, \ldots, X_m \in \mathbb{R}^{n \times n}$ be independent random PSD matrices such that $\lambda_{\max}(X_i) \leq L$ for each $i \in [m]$. For $X = \sum_{i=1}^{m} X_i$, $\mu_{\max} = \lambda_{\max}(\mathbb{E}[X])$, and $t > 0$,*

$$\mathbb{P}(\lambda_{\max}(X) \geq t\mu_{\max}) \leq n \left(\frac{e}{t}\right)^{t\mu_{\max}/L}.$$

The dimension factor of $n$ in Theorem 2.1 can essentially be replaced with a factor of the intrinsic dimension of the matrix $\mathbb{E}[X]$ relative to $\lambda_{\max}(\mathbb{E}[X])$. We only need to include an additional factor of 2 and enforce $t \geq L/\mu_{\max} + 1$:

**Theorem 2.2** ((Tropp et al., 2015), Theorem 7.2.1). *Let $X_1, \ldots, X_m \in \mathbb{R}^{n \times n}$ be independent random PSD matrices such that $\lambda_{\max}(X_i) \leq L$ for each $i \in [m]$. Also,*

*let $V \in S_+^n$ such that $V \succeq \mathbb{E}[X]$. For $X = \sum_{i=1}^{m} X_i$, $\mu_{\max} = \lambda_{\max}(V)$, and $t \geq L/\mu_{\max} + 1$,*

$$\mathbb{P}(\lambda_{\max}(X) \geq t\mu_{\max}) \leq 2 \operatorname{intdim}(V, \mu_{\max}) \left(\frac{e}{t}\right)^{t\mu_{\max}/L}.$$

*Remark* 2.3. We define the intrinsic dimension of a matrix relative to a scalar value, but Tropp defines intrinsic dimension of a PSD matrix $V$ to be equal to the quantity $\mathrm{tr}(V)/\lambda_{\max}(V)$. Our definition is more general and provides an an upper bound on Tropp's intrinsic dimension. The upper bound is convenient for lower bounding $x^{\top} X x$ over $x \in S(\mathbb{E}[X], \alpha)$, rather than just upper bounding $x^{\top} X x$ over $x \in S^{n-1}$. $\qquad\square$

Next, consider the following matrix Chernoff inequality for the minimum eigenvalue of a sum of independent random PSD matrices that are surely bounded.

**Theorem 2.4** ((Tropp et al., 2015), Theorem 5.1.1). *Let $X_1, \ldots, X_m \in \mathbb{R}^{n \times n}$ be independent random PSD matrices such that $\lambda_{\max}(X_i) \leq L$ for each $i \in [m]$. For $X = \sum_{i=1}^{m} X_i$, $\mu_{\min} = \lambda_{\min}(\mathbb{E}[X])$, and $\epsilon \in (0, 1]$,*

$$\mathbb{P}(\lambda_{\min}(X) \leq (1 - \epsilon)\mu_{\min}) \leq n e^{-\epsilon^2 \mu_{\min}/(2L)}.$$

Unlike Theorem 2.1, an intrinsic dimension analog has yet to be developed for Theorem 2.4. As pointed out in (Tropp et al., 2015), the techniques employed to establish Theorem 2.2 do not seem to apply in the context of the minimum eigenvalue.

It is also worthwhile to note that we cannot upper bound (6) with Theorem 2.4. It there is a direction $x$ per which $x^{\top} \mathbb{E}[X] x = 0$, and hence $\lambda_{\min}(\mathbb{E}[X]) = 0$, then Theorem 2.4 provides us with a trivial guarantee. The existence of such a direction, however, should not impact upper bounding (6). One might hope that we can directly apply Theorem 2.4 to the set $S(\mathbb{E}[X], \alpha)$, but the set is not a subspace (as seen in Figure 1). Although, an idea along these lines is exploited in the proof of Theorem 3.1; see Lemma 3.2.

### 2.2. One-sided approximation inequality

Let $X_1, \ldots, X_m$ be independent random PSD matrices such that $\lambda_{\max}(X_i) \leq L$ for each $i \in [m]$, $X = \sum_{i=1}^{m} X_i$, $\mu_{\min} = \lambda_{\min}(\mathbb{E}[X])$, and $\epsilon \in (0, 1]$. Theorem 2.4 enables us to ensure that

$$\min_{x \in S^{n-1}} x^{\top} X x = \lambda_{\min}(X)$$
$$> (1 - \epsilon)\lambda_{\min}(\mathbb{E}[X])$$
$$= (1 - \epsilon) \min_{x \in S^{n-1}} x^{\top} \mathbb{E}[X] x$$

with some amount of probability. In this subsection we show that we can more strongly ensure that

$$x^{\top} X x > (1 - \epsilon) x^{\top} \mathbb{E}[X] x, \quad \forall x \in S^{n-1}$$
$$\iff X \succ (1 - \epsilon)\mathbb{E}[X]$$

with the same amount of probability. That is, we show that we can ensure that $X$ is a *one-sided approximation* of $\mathbb{E}[X]$. A precise statement of the result that we establish is as follows.

**Theorem 2.5.** *Let $X_1, \ldots, X_m \in \mathbb{R}^{n \times n}$ be independent random PSD matrices such that $\lambda_{\max}(X_i) \leq L$ for each $i \in [m]$. For $X = \sum_{i=1}^m X_i$, $\mu_{\min} = \lambda_{\min}(\mathbb{E}[X])$, and $\epsilon \in (0,1]$,*

$$\mathbb{P}(X \not\succeq (1-\epsilon)\mathbb{E}[X])$$
$$= \mathbb{P}(\exists x \in \mathbb{R}^n \setminus \{0\} : x^\top X x \leq (1-\epsilon)x^\top \mathbb{E}[X]x)$$
$$\leq n e^{-\epsilon^2 \mu_{\min}/(2L)}.$$

*Proof.* See Appendix B.1. □

In our proof of Theorem 2.5, we first construct transformations $Y_i$, $i \in [m]$, of the matrices $X_i$, $i \in [m]$, respectively, with the property that $\mathbb{E}[\sum_{i=1}^m Y_i] = I$. Then, we apply Theorem 2.4 to the transformed matrices, and the desired result follows. So, Theorem 2.4 implies Theorem 2.5, but clearly Theorem 2.5 implies Theorem 2.4, so Theorem 2.5 is best thought of as a reformulation of Theorem 2.4.

Ultimately, we need Theorem 2.5 (instead of Theorem 2.4) in Section 3 to derive the intrinsic dimension concentration inequality. We direct the reader to Remark 3.4 at the end of Section 3 for an explanation.

## 3. Intrinsic dimension concentration inequality

We provide a statement of the intrinsic dimension concentration inequality that we develop in Theorem 3.1 below.

**Theorem 3.1.** *Let $X_1, \ldots, X_m \in \mathbb{R}^{n \times n}$ be independent random PSD matrices. Suppose that $\lambda_{\max}(X_i) \leq L$ for each $i \in [m]$. For $X = \sum_{i=1}^m X_i$, $\epsilon \in (0,1]$, and $16e^2 L/\epsilon^2 \leq \alpha \leq \lambda_{\max}(\mathbb{E}[X])$,*

$$\mathbb{P}(\exists x \in S(\mathbb{E}[X], \alpha) : x^\top X x \leq (1-\epsilon)\alpha)$$
$$\leq 48e^2 \operatorname{intdim}(\mathbb{E}[X], \epsilon^2 \alpha) e^{-\epsilon^4 \alpha/(512e^2 L)}.$$

*Proof.* See Appendix B.4. □

Theorem 3.1 tells us that we can guarantee that the quadratic form $x^\top X x$ is larger on a smaller set of directions $x \in S(\mathbb{E}[X], \alpha)$ per which the quadratic form $x^\top \mathbb{E}[X]x$ is larger. Accordingly, there is a tradeoff: For larger values of $\alpha$, we can guarantee $x^\top X x$ is larger, albeit in a smaller number of directions $x$. When we apply Theorem 3.1 in Section 4 towards developing an approximation guarantee for simple randomized rounding, we choose $\alpha$ to optimize this tradeoff. On a related note, observe that Theorem 3.1 does not preserve dependence on $\epsilon$ like Theorem 2.2. In our approximation analysis of simple randomized rounding, we take $\epsilon$

to be a constant, in which case the theorems provide similar guarantees.

Consider the extreme case in which $\alpha = \lambda_{\min}(\mathbb{E}[X])$, which is still possible under the assumptions of Theorem 3.1. Using the fact that $\operatorname{tr}(\mathbb{E}[X]) \geq n\lambda_{\min}(\mathbb{E}[X])$, it then follows that Theorem 3.1 provides a weaker guarantee than Theorem 2.4. Thus, as mentioned at the end of Subsection 1.1, the result is non-trivial when $\alpha > \lambda_{\min}(\mathbb{E}[X])$.

The remainder of this section is dedicated to outlining the proof of Theorem 3.1. First, we establish preliminaries and then, we present the proof outline.

**Proof preliminaries.** Let $X_1, \ldots, X_m \in \mathbb{R}^{n \times n}$ be independent random PSD matrices such that $\lambda_{\max}(X_i) \leq L$ for each $i \in [m]$. Also, let $X = \sum_{i=1}^m X_i$.

Let $u_1, \ldots, u_n$ be orthonormal eigenvectors of $\mathbb{E}[X]$, and let $\hat{\lambda}_1, \ldots, \hat{\lambda}_n$ denote the corresponding respective eigenvalues. That is, $\mathbb{E}[X]u_i = \hat{\lambda}_i u_i$ for each $i \in [n]$. We collect the eigenvectors into the columns of the orthonormal matrix $U := [u_1 | \cdots | u_n]$, and we collect the eigenvalues into the diagonal of the diagonal matrix $\Lambda := \operatorname{diag}(\hat{\lambda}_1, \ldots, \hat{\lambda}_n)$. The spectral decomposition of $\mathbb{E}[X]$ is then given by $\mathbb{E}[X] = U\Lambda U^\top$.

For $\beta > 0$, define the sets

$$I_1(\beta) := \{i \in [n] : \hat{\lambda}_i \geq \beta\},$$

and

$$I_2(\beta) := [n] \setminus I_1(\beta) = \{i \in [n] : \hat{\lambda}_i < \beta\},$$

which partition $[n]$ based on the size of the eigenvalues $\hat{\lambda}_1, \ldots, \hat{\lambda}_n$. Note that $I_1(\beta) \neq \emptyset$ if $\beta \leq \lambda_{\max}(\mathbb{E}[X])$. Similarly, $I_2(\beta) \neq \emptyset$ if $\beta > \lambda_{\min}(\mathbb{E}[X])$, but we do not exploit this fact.

For $\beta > 0$ and $j \in [2]$ such that $I_j(\beta) \neq \emptyset$, we collect the eigenvectors $u_i$, $i \in I_j(\beta)$ into the columns of the matrix $U_j(\beta)$. Then, we collect the eigenvalues $\hat{\lambda}_i$, $i \in I_j(\beta)$, into the diagonal of the diagonal matrix $\Lambda_j(\beta)$, and define

$$L_j(\beta) := \{U_j(\beta)\pi : \pi \in \mathbb{R}^{|I_j(\beta)|}\}$$

to be the subspace spanned by the eigenvectors $u_i, i \in I_j(\beta)$. Next, let

$$S_j(\beta) := S^{n-1} \cap L_j(\beta) = \{U_j(\beta)\pi : \pi \in S^{|I_j(\beta)|-1}\}$$

to be the set of points with unit Euclidean norm that lie within the subspace $L_j(\beta)$, and we define the independent random PSD matrices

$$Y_{ij} := U_j(\beta)^\top X_i U_j(\beta), \quad (i,j) \in [m] \times [2]$$

as well as $Y_j := \sum_{i=1}^m Y_{ij} = U_j(\beta)^\top X U_j(\beta)$.

Note that each matrix $Y_{ij}$ is a $|I_j(\beta)| \times |I_j(\beta)|$ matrix. Also, note that

$$\begin{aligned}
E[Y_j] &= U_j(\beta)^\top \mathbb{E}[X] U_j(\beta) \\
&= U_j(\beta)^\top U \Lambda U^\top U_j(\beta) \qquad (7) \\
&= \Lambda_j,
\end{aligned}$$

and that for each $i \in [m]$,

$$\begin{aligned}
\lambda_{\max}(Y_{ij}) &= \lambda_{\max}(U_j(\beta)^\top X_i U_j(\beta)) \\
&\leq \lambda_{\max}(X_i)\lambda_{\max}(U_j(\beta)^\top U_j(\beta)) \leq L,
\end{aligned} \qquad (8)$$

where the first inequality follows from the same argument provided in the proof of Theorem 2.5 in Appendix B.1 (specifically, see Lemma B.2); the second inequality follows from $\lambda_{\max}(X_i) \leq L$ together with $\lambda_{\max}(U_j(\beta)^\top U_j(\beta)) = \lambda_{\max}(I) = 1$. Below we apply Theorem 2.5 and 2.2 with respect to the matrices $Y_{i1}$, $i \in [m]$ and $Y_{i2}$, $i \in [m]$, respectively, and use the facts (7) and (8) in the process.

**Proof outline.** For larger values of $\beta > 0$, the dimension of the subspace $L_1(\beta)$ is smaller (or at least no larger). Accordingly, applying Theorem 2.5 with respect to matrices $Y_{i1}$, $i \in [m]$, which capture the behavior of the matrices $X_i$, $i \in [m]$, on the subspace $L_1(\beta)$, we should be able to guarantee a better one-sided approximation on $L_1(\beta)$ for larger values of $\beta$, as captured by the factor of $\mathrm{intdim}(\mathbb{E}[X], \beta)$ in Lemma 3.2 below. This intrinsic dimension factor appears (as opposed to $n$) because the dimension of $L_1(\beta)$ is bounded above by it.

**Lemma 3.2.** *For $\beta > 0$ such that $I_1(\beta) \neq \emptyset$, and $\delta \in (0, 1]$, it holds that*

$$\begin{aligned}
\mathbb{P}(\exists x &\in L_1(\beta) \setminus \{0\} : x^\top X x \leq (1-\delta)x^\top \mathbb{E}[X]x) \\
&\leq \mathrm{intdim}(\mathbb{E}[X], \beta)e^{-\delta^2 \beta/(2L)}.
\end{aligned}$$

*Proof.* See Appendix B.2. □

Lemma 3.2 provides us with a one-sided approximation guarantee for the directions $S_1(\beta)$, while Theorem 3.1 states a weaker guarantee for the larger set of directions $S_1(\mathbb{E}[X], \beta) \supseteq S_1(\beta)$. Accordingly, we work towards showing that Lemma 3.2 implies the weaker guarantee for the directions $S_1(\mathbb{E}[X], \beta) \setminus S_1(\beta)$.

We are only missing the following ingredient. Applying Theorem 2.2 to the matrices $Y_{i2}, i \in [m]$, which capture the behavior of the matrices $X_i$, $i \in [m]$ with respect to the directions $S_2(\beta)$, we can ensure that the quadratic form $x^\top X x$ is not too large in each direction $x \in S_2(\beta)$. Because Theorem 2.2 provides an intrinsic dimension inequality, we obtain an intrinsic dimension inequality:

**Lemma 3.3.** *For $\beta > 0$ such that $I_1(\beta), I_2(\beta) \neq \emptyset$, and $t \geq \max\{L/\beta + 1, e^2\}$,*

$$\mathbb{P}\left(\max_{x \in S_2(\beta)} x^\top X x \geq t\beta\right) \leq 2\,\mathrm{intdim}(\mathbb{E}[X], \beta)e^{-t\beta/L}.$$

*Proof.* See Appendix B.3. □

A sketch of the proof of Theorem 3.1 is as follows. The aim is to ensure that $x^\top X x > (1-\epsilon)\alpha$ for all $x \in S(\mathbb{E}[X], \alpha)$ with some amount of probability. Fix $x \in S(\mathbb{E}[X], \alpha)$. First, we carefully specify a value of $\beta > 0$ (details behind the specification to come). Assume that $I_j(\beta) \neq \emptyset$ for each $j \in [2]$. (The proof otherwise ends up being straightforward.) Because $L_1(\beta)$ is the orthogonal complement of $L_2(\beta)$, we can decompose $x \in S(\mathbb{E}[X], \beta)$ as $x = x_1 + x_2$, where $x_1 \in L_1(\beta)$ and $x_2 \in L_2(\beta)$. Because $X$ is a random PSD matrix, we can factor $X$ as $X = X^{1/2}X^{1/2}$, where $X^{1/2}$ is a random matrix. From $x = x_1 + x_2$ and $X = X^{1/2}X^{1/2}$, we have that

$$\begin{aligned}
x^\top X x &= x_1^\top X x_1 + 2x_1^\top X^{1/2}X^{1/2}x_2 + x_2^\top X x_2 \\
&\geq x_1^\top X x_1 - 2\sqrt{x_1^\top X x_1}\sqrt{x_2^\top X x_2} + x_2^\top X x_2 \\
&= \left(\sqrt{x_1^\top X x_1} - \sqrt{x_2^\top X x_2}\right)^2, \qquad (9)
\end{aligned}$$

where the inequality follows from the Cauchy-Schwarz inequality. Taking $\beta$ to be sufficiently small, we can ensure that $x_2^\top X x_2$ is sufficiently small by Lemma 3.3, and we can ensure that $x_1^\top X x_1$ is sufficiently large by Lemma 3.2.

*Remark* 3.4. To ensure that $x_1^\top X x_1$ is sufficiently large, it seems critical that we use the one-sided approximation $x_1^\top X x_1 > (1-\epsilon)x_1^\top \mathbb{E}[X]x_1$ provided by Lemma 3.2 (which ultimately traces back to Theorem 2.5), rather than a weaker lower bound of the form $x_1^\top X x_1/\|x_1\|^2 > (1-\epsilon)\beta$, which could be obtained from Theorem 2.4. When $\|x_1\|$ is small, the weaker bound is not helpful, but we can regardless apply the one-sided approximation as long as $x_1^\top \mathbb{E}[X]x_1$ is sufficiently large. □

# 4. Approximation guarantee for simple randomized rounding

We present our main approximation result for simple randomized rounding in Theorem 4.1 below. The theorem states that simple randomized rounding provides a constant-factor approximation if $\mathrm{INC} = \Omega(R \ln k)$, where recall that where $R := \max_{i \in [m]} \mathrm{tr}(A_i)$. Note that the theorem does not account for the fact that $z_{\mathrm{round}}$ should be feasible for (1), i.e., satisfy $\|z_{\mathrm{round}}\|_0 \leq k$; we address this issue further below. Also, note that the constants specified in the theorem are chosen to make the verification of the proof more convenient. For instance, the constant in the lower bound on INC could be increased to obtain a better approximation factor.

**Theorem 4.1.** *Suppose that $k \geq 2$. If $\mathrm{INC} \geq 2^{14}e^2 R \ln k$, then*

$$\mathbb{P}\Big(\lambda_{\min}\Big(M + \sum_{i=1}^{m}[z_{\mathrm{round}}]_i A_i\Big) \geq (1/4)\mathrm{OPT}\Big) \geq 61/64.$$

*Proof.* See Appendix B.5. □

Our proof of Theorem 4.1 utilizes Theorem 3.1 and follows the argument outlined in Subsection 1.1. Recall that $W := \max_{i\in[m]} \lambda_{\max}(A_i)$. Following a similar proof, one could instead apply the matrix Chernoff inequality provided in Theorem 2.4 to obtain the condition $\mathrm{INC} = \Omega(W \ln n)$, which unlike our guarantee, does not shed light on the $k < n$ setting. However, the condition $\mathrm{INC} = \Omega(W \ln n)$ does depend on $W \leq R$. If the augmentation matrices are rank-one matrices, then $R = W$, and there is no advantage to this alternative condition.

To further motivate Theorem 4.1, it is worthwhile to note that it is possible that $\mathrm{INC} = Rk$. Consider the following instance of the max-min eigenvalue augmentation problem (1). Suppose that the augmentation matrices are rank-one matrices, in which case $R = W$. Further suppose that $M = Wk(I - e_1 e_1^\top)$, where $e_1$ is the 1st standard unit vector. Finally suppose that there are $k$ (amongst the $m$) augmentation matrices that are all equal to $We_1 e_1^\top$. Then, it is clearly optimal to select these $k$ matrices. It follows that $\mathrm{INC} = Wk$, which for sufficiently large $k$ implies that $\mathrm{INC} \geq 2^{14}e^2 R \ln k$, from which we can conclude from Theorem 4.1 that simple randomized rounding provides a constant-factor approximation. However, no matter how large we take $k$, we can always take $n$ such that $k << \ln n$, in which case $\mathrm{INC} = Wk$ does not imply that $\mathrm{INC} = \Omega(W \ln n)$.

**Feasibility of $z_{\mathrm{round}}$.** To establish a guarantee that additionally ensures $z_{\mathrm{round}}$ is feasible, we make use of Proposition 4.2 below. (Curiously, we were not able to find a statement/proof of this fact in a more recent reference.)

**Proposition 4.2** ((Jogdeo & Samuels, 1968), Theorem 3.2). *Suppose that the mean $\mu = \mathbb{E}[X]$ of a Poisson binomial random variable $X$ is an integer. Then, $\mu$ is the median of $X$.*

Observe that the quantity $\|z_{\mathrm{round}}\|_0$ is a Poisson binomial random variable with mean $\|z_{\mathrm{sdp}}\|_1 \leq k$. It follows that

$$\mathbb{P}(\|z_{\mathrm{round}}\|_0 \leq k) \geq 1/2. \tag{10}$$

Indeed, if $\|z_{\mathrm{sdp}}\| = k$, then (10) follows immediately from Proposition 4.2. Suppose that $\|z_{\mathrm{sdp}}\| < k$ (which is possible, as we only require $(z_{\mathrm{sdp}}, \eta_{\mathrm{sdp}})$ be optimal for the relaxation (2)). For a Poisson binomial random variable $Z$ that has mean $k - \|z_{\mathrm{sdp}}\|_1$ and that is independent of $\|z_{\mathrm{round}}\|_0$,

we have that $\mathbb{P}(\|z_{\mathrm{round}}\|_0 \leq k) \geq \mathbb{P}(\|z_{\mathrm{round}}\|_0 + Z \leq k) \geq 1/2$, where the second inequality follows from the fact that $\|z_{\mathrm{round}}\|_0 + Z$ is a Poisson binomial random variable with mean $k$.

With (10) in hand, we obtain the following corollary to Theorem 4.1:

**Corollary 4.3.** *Suppose that $k \geq 2$. If $\mathrm{INC} \geq 2^{14}e^2 R \ln k$, then*

$$\mathbb{P}\Big(\lambda_{\min}\Big(M + \sum_{i=1}^{m}[z_{\mathrm{round}}]_i A_i\Big) \geq (1/4)\mathrm{OPT}, \|z\|_0 \leq k\Big)$$
$$\geq 29/64.$$

## 5. Discussion

We considered the max-min eigenvalue augmentation problem (1) and focused on the setting in which $k < n$. We showed that simple randomized rounding provides a constant-factor approximation if the *optimal increase* is sufficiently large, specifically if $\mathrm{OPT} - \lambda_{\min}(M) = \Omega(R \ln k)$; see Theorem 4.1. To establish the guarantee, we derived a matrix concentration inequality that is of independent interest; see Theorem 3.1. The inequality can be interpreted as an *intrinsic dimension* analog of the matrix Chernoff inequality for the minimum eigenvalue of a sum of independent random PSD matrices.

There are a number of directions that we aim to pursue for future work. First, we aim to investigate the extent to which we can improve the dependence on $\epsilon$ in Theorem 3.1. It is also of interest to determine the extent to which Theorem 4.1 is tight. One would imagine that it is tight (up to a constant factor) given that Theorem 2.4 is tight; see (Tropp et al., 2015). Finally, we plan to explore the empirical approximation performance of simple randomized rounding through a computational study.

## Acknowledgments

This work was partially supported by the ONR grant N00014-22-1-2674. The authors would like to thank the anonymous referees for their helpful and constructive comments that allowed us to improve this paper.

## Impact Statement

This paper presents work whose goal is to advance the field of Machine Learning. There are many potential societal consequences of our work, none which we feel must be specifically highlighted here.

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

## A. Omitted proofs

## B. Algebraic connectivity as a minimum eigenvalue

**Lemma B.1.** *Let $G$ be a simple undirected graph. Then, $\lambda_2(L) = \lambda_{\min}(L + \mathbf{1}\mathbf{1}^\top)$, where $L$ is the Laplacian of $G$.*

*Proof.* Because $L$ is positive semidefinite and $L\mathbf{1} = 0$, it holds that $\mathbf{1}$ is an eigenvector of $L$ with eigenvalue $\lambda_{\min}(L) = 0$. We consider two cases:

**Case 1.** Suppose that $G$ is disconnected. Then $\lambda_2(L) = 0$, and it follows that there is an eigenvector $u$ of $L$ that has eigenvalue $0$ and is orthogonal to $\mathbf{1}$. Hence $(L + \mathbf{1}\mathbf{1}^\top)u = 0$, implying that $\lambda_{\min}(L + \mathbf{1}\mathbf{1}^\top) = 0$, as $L + \mathbf{1}\mathbf{1}^\top$ is positive semidefinite. Thus $\lambda_2(L) = 0 = \lambda_{\min}(L + \mathbf{1}\mathbf{1}^\top)$.

**Case 2.** Suppose that $G$ is connected. Hence $\lambda_2(L) > 0$. Let $u_1, \ldots, u_n$ be orthonormal eigenvectors of $L$ with eigenvalues $\lambda_1(L) \leq \cdots \leq \lambda_n(L)$, respectively. Because $\lambda_{\min}(L) = 0 < \lambda_2(L)$, it follows that $u_1 = \mathbf{1}$.

Note that $(L + \mathbf{1}\mathbf{1}^\top)\mathbf{1} = n\mathbf{1}$, and $(L + \mathbf{1}\mathbf{1}^\top)u_i = \lambda_i(L)u_i$ for each $2 \leq i \leq n$. As a result, the eigenvalues of $L + \mathbf{1}\mathbf{1}^\top$ are $n, \lambda_2(L), \ldots, \lambda_n(L)$. Thus, the desired result follows from the fact that $\lambda_2(L) \leq n$. □

### B.1. Proof of Theorem 2.5

If $\mu_{\min} = 0$, then the theorem trivially holds. Hence, suppose that $\mu_{\min} > 0$. Let $Q = \mathbb{E}[X]$. Because $\lambda_{\min}(Q) = \mu_{\min} > 0$, the matrix $Q$ is positive definite. Accordingly, we can factor $Q$ as $Q = Q^{1/2}Q^{1/2}$, where $Q^{1/2}$ is an $n \times n$ invertible matrix. Define the random PSD matrices $Y_i := Q^{-1/2}X_iQ^{-1/2}$, $i \in [m]$. Also, define $Y := \sum_{i=1}^m Y_i = Q^{-1/2}XQ^{-1/2}$.

Our proof of Theorem 2.5 follows from applying Theorem 2.4 to the matrices $Y_i$, $i \in [m]$. Towards this end, note that $\mathbb{E}[Y] = Q^{-1/2}\mathbb{E}[X]Q^{-1/2} = I$, and hence, $\lambda_{\min}(\mathbb{E}[Y]) = 1$. Also, make note of the observations captured in Lemmas B.2 and B.3 below.

**Lemma B.2.** $\lambda_{\max}(Y_i) \leq L/\mu_{\min}$ *for each* $i \in [m]$.

*Proof.* Note that for each $i \in [m]$ we have that:

$$
\begin{aligned}
\lambda_{\max}(Y_i) &= \max_{x \in S^{n-1}} x^\top Y_i x \\
&= \max_{x \in S^{n-1}} x^\top Q^{-1/2}X_iQ^{-1/2}x \\
&\leq \left( \max_{x \in S^{n-1}} \frac{x^\top Q^{-1/2}X_iQ^{-1/2}x}{\|Q^{-1/2}x\|^2} \right) \left( \max_{x \in S^{n-1}} \|Q^{-1/2}x\|^2 \right) \\
&= \left( \max_{y \in S^{n-1}} y^\top X_i y \right) \left( \max_{x \in S^{n-1}} x^\top Q^{-1}x \right) \\
&= \lambda_{\max}(X_i)\lambda_{\max}(Q^{-1}) \\
&\leq L/\mu_{\min},
\end{aligned}
$$

where the third equality follows from the change of variables $y = Q^{-1}x/\|Q^{-1/2}x\|$, and the last inequality follows from $\lambda_{\max}(X_i) \leq L$ and $\lambda_{\max}(Q^{-1}) = 1/\lambda_{\min}(Q) = 1/\mu_{\min}$. □

**Lemma B.3.** $\{X \not\succ (1 - \epsilon)\mathbb{E}[X]\} = \{Y \not\succ (1 - \epsilon)I\}$.

*Proof.* It is sufficient to show that $\{X \succ (1 - \epsilon)\mathbb{E}[X]\} = \{Y \succ (1 - \epsilon)I\}$. Suppose that $X \succ (1 - \epsilon)\mathbb{E}[X]$. For $x \in S^{n-1}$, we have that

$$x^\top Yx = x^\top Q^{-1/2}XQ^{-1/2}x > (1 - \epsilon)x^\top Q^{-1/2}\mathbb{E}[X]Q^{-1/2}x = (1 - \epsilon)x^\top Ix,$$

where the inequality follows from $X \succ (1 - \epsilon)\mathbb{E}[X]$. Thus, $Y \succ (1 - \epsilon)I$.

Suppose that $Y \succ (1 - \epsilon)I$. For $x \in S^{n-1}$, we have that

$$x^\top Xx = x^\top Q^{1/2}YQ^{1/2}x > (1 - \epsilon)x^\top Qx = (1 - \epsilon)x^\top \mathbb{E}[X]x,$$

where the inequality follows from $Y \succ (1 - \epsilon)I$. Thus, $X \succ (1 - \epsilon)\mathbb{E}[X]$. □

We are now prepared to prove Theorem 2.5:

*Theorem 2.5.* From Lemma B.3, we have that

$$\mathbb{P}(X \not\succeq (1-\epsilon)\mathbb{E}[X]) = \mathbb{P}(Y \not\succeq (1-\epsilon)I) = \mathbb{P}(\lambda_{\min}(Y) \leq 1-\epsilon) \leq ne^{-\epsilon^2\mu_{\min}/(2L)},$$

where the inequality follows from Theorem 2.4, which can be applied due to Lemma B.2. $\qquad\square$

## B.2. Proof of Lemma 3.2

*Lemma 3.2.* For notational convenience, we suppress dependence on $\beta$ and write $I_1(\beta)$, $L_1(\beta)$, $S_1(\beta)$, $U_1(\beta)$, and $\Lambda_1(\beta)$ as $I_1$, $L_1$, $S_1$, $U_1$ and $\Lambda_1$, respectively.

First, note that $\text{tr}(\mathbb{E}[X]) = \sum_{i=1}^n \hat{\lambda}_i \geq \sum_{i \in I_1} \hat{\lambda}_i \geq |I_1|\beta$, where the second inequality follows from the definition of $I_1$, and hence:

$$|I_1| \leq \frac{\text{tr}(\mathbb{E}[X])}{\beta} = \text{intdim}(\mathbb{E}[X], \beta). \tag{11}$$

From the definition of $L_1$, we have the following sequence:

$$\begin{aligned}
&\mathbb{P}(\exists x \in L_1 \setminus \{0\} : x^\top X x \leq (1-\delta)x^\top \mathbb{E}[X]x) \\
&= \mathbb{P}(\exists \pi \in \mathbb{R}^{|I_1|-1} \setminus \{0\} : \pi^\top U_1^\top X U_1 \pi \leq (1-\delta)\pi^\top U_1^\top \mathbb{E}[X]U_1\pi) \\
&= \mathbb{P}(\exists \pi \in \mathbb{R}^{|I_1|-1} \setminus \{0\} : \pi^\top Y_1 \pi \leq (1-\delta)\pi^\top \Lambda_1\pi) \\
&\leq |I_1|e^{-\delta^2\lambda_{\min}(\Lambda_1)/(2L)} \\
&\leq \text{intdim}(\mathbb{E}[X], \beta)e^{-\delta^2 \min_{i \in I_1} \hat{\lambda}_i/(2L)} \\
&\leq \text{intdim}(\mathbb{E}[X], \beta)e^{-\delta^2\beta/(2L)},
\end{aligned}$$

where the second equality follows from $U_1^\top \mathbb{E}[X]U_1 = U_1^\top U\Lambda U^\top U_1 = \Lambda_1$; the first inequality follows from applying Theorem 2.5 with respect to the matrices $Y_{i1}$, $i \in [m]$; the second inequality follows from (11) and $\lambda_{\min}(\Lambda_1) = \min_{i \in I_1} \hat{\lambda}_i$; and the last inequality follows from the fact that $\hat{\lambda}_i \geq \beta$ for $i \in I_1$. $\qquad\square$

## B.3. Proof of Lemma 3.3

*Lemma 3.3.* For notational convenience, we suppress dependence on $\beta$ in the same spirit as the proof of Lemma 3.2.

Take any index $j \in I_1$, and let $V = \mathbb{E}[Y_2] + \beta u_j u_j^\top$. We introduce $V$ to apply Theorem 2.2 with $V$. Note that $V \succeq \mathbb{E}[Y_2]$ because $\beta > 0$. Also, note that the eigenvalues of $V$ are given by $\hat{\lambda}_i$, $i \in I_2$, and $\beta$, so $\lambda_{\max}(V) = \beta$ because $\hat{\lambda}_i < \beta$ for each $i \in I_2$ by definition. Finally, note that

$$\text{tr}(V) = \text{tr}(\mathbb{E}[Y_2]) + \beta = \text{tr}(\Lambda_2) + \beta = \sum_{i \in I_2} \hat{\lambda}_i + \beta \leq \sum_{i=1}^n \hat{\lambda}_i = \text{tr}(\mathbb{E}[X]), \tag{12}$$

where the inequality follows from $\beta \leq \lambda_j$ for $j \in I_1$. From the definition of $S_2$,

$$\begin{aligned}
\mathbb{P}\left(\max_{x \in S_2} x^\top X x \geq t\beta\right) &= \mathbb{P}\left(\max_{\pi \in S^{|I_2|-1}} \pi^\top U_2^\top X U_2 \pi \geq t\beta\right) \\
&= \mathbb{P}\left(\lambda_{\max}(U_2^\top X U_2) \geq t\beta\right) \\
&= \mathbb{P}\left(\lambda_{\max}(Y_2) \geq t\lambda_{\max}(V)\right) \\
&\leq 2\,\text{intdim}(V, \lambda_{\max}(V))\left(\frac{e}{t}\right)^{t\lambda_{\max}(V)/L} \\
&= 2\frac{\text{tr}(V)}{\lambda_{\max}(V)}\left(\frac{e}{t}\right)^{t\lambda_{\max}(V)/L} \\
&\leq 2\frac{\text{tr}(\mathbb{E}[X])}{\beta}\left(\frac{e}{t}\right)^{t\beta/L} \\
&\leq 2\,\text{intdim}(\mathbb{E}[X], \beta)e^{-t\beta/L},
\end{aligned}$$

where the first inequality follows from Theorem 2.2 with respect to $Y_{i2}$, $i \in [m]$ (here we use the supposition that $t \geq L/\beta + 1 = L/\lambda_{\max}(V) + 1$ by $\lambda_{\max}(V) = \beta$); the second inequality follows from (12) and $\lambda_{\max}(V) = \beta$; and the last inequality follows from $t \geq e^2$. $\qquad\square$

## B.4. Proof of Theorem 3.1

*Theorem 3.1.* Let $\beta = \epsilon^2 \alpha / (16e^2)$. For notational convenience, we suppress dependence on $\beta$ and for each $j \in [2]$ write $I_j(\beta)$, $L_j(\beta)$, $S_j(\beta)$, $U_j(\beta)$, and $\Lambda_j(\beta)$ as $I_j$, $L_j$, $S_j$, $U_j$ and $\Lambda_j$, respectively.

First, note that because $\beta < \alpha \leq \lambda_{\max}(X)$, it holds that $I_1 \neq \emptyset$. Thus, from Lemma 3.2 with $\delta = \epsilon/4$, we have that:

$$\mathbb{P}(\exists x \in L_1 \setminus \{0\} : x^\top X x \leq (1 - \epsilon/4) x^\top \mathbb{E}[X] x)$$
$$\leq \mathrm{intdim}(\mathbb{E}[X], \beta) e^{-\epsilon^2 \beta / (32L)}$$
$$= 16e^2 \, \mathrm{intdim}(\mathbb{E}[X], \epsilon^2 \alpha) e^{-\epsilon^4 \alpha / (512 e^2 L)}, \tag{13}$$

where the equality follows from $\beta = \epsilon^2 \alpha / (16e^2)$.

Suppose that $S_2 = \emptyset$. Then, it holds that $S_1 = S^{n-1}$. From the definition of $S(\mathbb{E}[X], \alpha)$,

$$\mathbb{P}(\exists x \in S(\mathbb{E}[X], \alpha) : x^\top X x \leq (1 - \epsilon)\alpha)$$
$$\leq \mathbb{P}(\exists x \in S(\mathbb{E}[X], \alpha) : x^\top X x \leq (1 - \epsilon) x^\top \mathbb{E}[X] x)$$
$$\leq \mathbb{P}(\exists x \in S_1 : x^\top X x \leq (1 - \epsilon) x^\top \mathbb{E}[X] x)$$
$$\leq \mathbb{P}(\exists x \in S_1 : x^\top X x \leq (1 - \epsilon/4) x^\top \mathbb{E}[X] x)$$
$$\leq 16e^2 \, \mathrm{intdim}(\mathbb{E}[X], \epsilon^2 \alpha) e^{-\epsilon^4 \alpha / (512 e^2 L)},$$

where the second inequality follows from $S_1 = S^{n-1} \supseteq S(\mathbb{E}[X], \alpha)$, and the last inequality follows from (13). Thus, the desired result follows, so let us assume that $S_2 \neq \emptyset$.

Then, note that $L/\beta = 16e^2 L / (\alpha \epsilon^2) \leq 1$ because $\alpha \geq 16e^2 L / \epsilon^2$ (recall the conditions in the theorem statement). Hence, we can apply Lemma 3.3 with $t = e^2$ to obtain that

$$\mathbb{P}\left(\max_{x \in S_2} x^\top X x \geq e^2 \beta\right) \leq 2 \, \mathrm{intdim}(\mathbb{E}[X], \beta) e^{-e^2 \beta / L}$$
$$= 32e^2 \, \mathrm{intdim}(\mathbb{E}[X], \epsilon^2 \alpha) e^{-\epsilon^2 \alpha / (16L)}, \tag{14}$$

where the equality follows from $\beta = \epsilon^2 \alpha / (16e^2)$. Accordingly, for the following conditions:

$$x^\top X x > (1 - \epsilon/4) x^\top \mathbb{E}[X] x, \quad \forall x \in L_1 \setminus \{0\} \tag{15}$$
$$x^\top X x < e^2 \beta, \qquad\qquad \forall x \in S_2, \tag{16}$$

we have from (13) and (14) that

$$\mathbb{P}((15) \text{ and } (16) \text{ hold})$$
$$\geq 1 - 16e^2 \, \mathrm{intdim}(\mathbb{E}[X], \epsilon^2 \alpha) e^{-\epsilon^4 \alpha / (512 e^2 L)} - 32e^2 \, \mathrm{intdim}(\mathbb{E}[X], \epsilon^2 \alpha) e^{-\epsilon^2 \alpha / (16L)}$$
$$\geq 1 - 48e^2 \, \mathrm{intdim}(\mathbb{E}[X], \epsilon^2 \alpha) \cdot e^{-\epsilon^4 \alpha / (512 e^2 L)},$$

where the second inequality follows from $e^{-\epsilon^2 \alpha / (16L)} \leq e^{-\epsilon^4 \alpha / (512 e^2 L)}$. Thus, taking $x \in S(\mathbb{E}[X], \alpha)$ and assuming that (15)-(16) hold, it is sufficient to show that

$$x^\top X x > (1 - \epsilon)\alpha. \tag{17}$$

Because $L_1$ is the orthogonal complement of $L_2$, there exists a point $x_1 \in L_1$ and point $x_2 \in L_2$ such that $x = x_1 + x_2$.

Because $x \in S(\mathbb{E}[X], \alpha)$,

$$
\begin{aligned}
\alpha &\leq x^\top \mathbb{E}[X] x \\
&= x^\top U \Lambda U^\top x \\
&= x^\top U_1 \Lambda_1 U_1^\top x + x^\top U_2 \Lambda_2 U_2^\top x \\
&\leq x^\top U_1 \Lambda_1 U_1^\top x + \lambda_{\max}(U_2 \Lambda_2 U_2^\top) \\
&= x^\top U_1 \Lambda_1 U_1^\top x + \max_{i \in I_2} \hat{\lambda}_i \\
&\leq x^\top U_1 \Lambda_1 U_1^\top x + \beta \\
&= x^\top U_1 \Lambda_1 U_1^\top x + \epsilon^2 \alpha/(16e^2),
\end{aligned} \tag{18}
$$

where the second inequality follows from $x \in S^{n-1}$; the third equality follows from the fact that the eigenvalues of $U_2 \Lambda_2 U_2^\top$ are $\hat{\lambda}_i$, $i \in I_2$; and the last inequality follows from the definition of $I_2$. Rearranging (18) gives

$$
\begin{aligned}
(1 - \epsilon^2/(16e^2))\alpha &\leq x^\top U_1 \Lambda_1 U_1^\top x \\
&= x_1^\top U_1 \Lambda_1 U_1^\top x_1 \\
&= x_1^\top U \Lambda U^\top x_1 \\
&= x_1^\top \mathbb{E}[X] x_1,
\end{aligned} \tag{19}
$$

where the first equality follows from $x = x_1 + x_2$ and $U_1^\top x_2 = 0$, and the second equality follows from $U_2^\top x_1 = 0$. From (19), we see that $x_1 \neq 0$, so $x_1 \in L_1 \setminus \{0\}$. Hence, from (15) we have that

$$
x_1^\top X x_1 > (1 - \epsilon/4) x_1^\top \mathbb{E}[X] x_1 \geq (1 - \epsilon/4)(1 - \epsilon^2/(16e^2))\alpha > (1 - \epsilon/4)^2 \alpha, \tag{20}
$$

where the second inequality follows from (19).

We claim that

$$
x_2^\top X x_2 < (\epsilon^2/16)\alpha. \tag{21}
$$

If $x_2 = 0$, then clearly (21) holds. So, suppose that $x_2 \neq 0$. Because $\|x_2\| \leq 1$ as well, we have that

$$
x_2^\top X x_2 \leq \frac{x_2^\top X x_2}{\|x_2\|^2} < e^2 \beta = (\epsilon^2/16)\alpha,
$$

where the second inequality follows from (16).

Finally, recall from (9) that

$$
\begin{aligned}
x^\top X x &\geq \left( \sqrt{x_1^\top X x_1} - \sqrt{x_2^\top X x_2} \right)^2 \\
&> (\sqrt{(1 - \epsilon/4)^2 \alpha} - \sqrt{(\epsilon^2/16)\alpha})^2 \\
&= (1 - \epsilon + \epsilon^2/4)\alpha \\
&> (1 - \epsilon)\alpha,
\end{aligned}
$$

where the second inequality follows from $x_1^\top X x_1 > (1 - \epsilon/4)^2 \alpha > (1/16)\epsilon^2 \alpha > x_2^\top X x_2$; recall (20) and (21). Thus, (17) holds, and the proof is complete. $\qquad \square$

### B.5. Proof of Theorem 4.1

*Theorem 4.1.* Let $C = 2^{14} e^2$, so we have INC $\geq CR \ln k$. We apply Theorem 3.1 with respect to the random PSD matrices $X_i = [z_{\text{round}}]_i A_i$, $i \in [m]$. We take $L = \max_{i \in [m]} \lambda_{\max}(A_i)$ (which is clearly a valid choice), $\epsilon = 1/2$, and $\alpha = (1/2) \text{INC}$. Verifying the final condition of the theorem, we have that

$$
16e^2 L/\epsilon^2 = 64e^2 L \leq 2^{14} e^2 L \leq \text{INC} \leq \lambda_{\max}\left( \sum_{i=1}^{m} [z_{\text{sdp}}]_i A_i \right),
$$

where the second inequality follows from $\text{INC} \geq 2^{14}e^2R\ln k \geq 64e^2L$ (recall $R \geq L$), and the last inequality follows from the fact that $\text{INC} = \lambda_{\min}(M + \sum_{i=1}^m [z_{\text{sdp}}]_i A_i) - \lambda_{\min}(M) \leq \lambda_{\max}(\sum_{i=1}^m [z_{\text{sdp}}]_i A_i)$.

Thus, the last condition of Theorem 3.1 holds, and we obtain that

$$
\mathbb{P}\Big( \exists x \in S\Big(\sum_{i=1}^m [z_{\text{sdp}}]_i A_i, (1/2)\text{INC}\Big) : x^\top\Big(\sum_{i=1}^m [z_{\text{round}}]_i A_i\Big)x \leq (1/4)\text{INC}\Big)
$$

$$
\leq 48e^2\,\text{intdim}\,\Big(\sum_{i=1}^m [z_{\text{sdp}}]_i A_i, (1/8)\text{INC}\Big)e^{-\text{INC}/(2^{14}e^2L)}
$$

$$
= 384e^2\,\text{intdim}\,\Big(\sum_{i=1}^m [z_{\text{sdp}}]_i A_i, \text{INC}\Big)e^{-\text{INC}/(2^{14}e^2L)}
$$

$$
\leq 384e^2\,\text{intdim}\,\Big(\sum_{i=1}^m [z_{\text{sdp}}]_i A_i, CR\ln k\Big)e^{-R\ln k/L}
$$

$$
= \frac{384}{2^{14}} \cdot \frac{\text{tr}(\sum_{i=1}^m [z_{\text{sdp}}]_i A_i)}{R\ln k} \cdot e^{-R\ln k/L}
$$

$$
\leq \frac{384}{2^{14}} \cdot \frac{k}{\ln k} \cdot e^{-\ln k}
$$

$$
= \frac{3}{128} \cdot \frac{1}{\ln k}
$$

$$
\leq \frac{3}{64} \tag{22}
$$

where the second inequality follows from the supposition $\text{INC} \geq CR\ln k$ and $C = 2^{14}e^2$; the third inequality follows from $\text{tr}(\sum_{i=1}^m [z_{\text{sdp}}]_i A_i) \leq R\sum_{i=1}^m [z_{\text{sdp}}]_i \leq Rk$ and $L \leq R$; and the last inequality follows from $\ln k \geq (1/2)$ (as $k \geq 2$).

From (22), it holds with probability at least $61/64$ that

$$
x^\top\Big(\sum_{i=1}^m [z_{\text{round}}]_i A_i\Big)x > (1/4)\text{INC} \quad \forall x \in S\Big(\sum_{i=1}^m [z_{\text{sdp}}]_i A_i, (1/2)\text{INC}\Big). \tag{23}
$$

Fix $x \in S^{n-1}$. Then, it is sufficient to show that

$$
x^\top\Big(M + \sum_{i=1}^m [z_{\text{round}}]_i A_i\Big)x \geq (1/4)\text{OPT} \tag{24}
$$

under the assumption that (23) holds.

**Case 2.1.** Suppose that $x \notin S(\sum_{i=1}^m [z_{\text{sdp}}]_i A_i, (1/4)\text{INC})$. Because $x \in S^{n-1}$,

$$
\lambda_{\min}\Big(M + \sum_{i=1}^m [z_{\text{sdp}}]_i A_i\Big) \leq x^\top\Big(M + \sum_{i=1}^m [z_{\text{sdp}}]_i A_i\Big)x < x^\top Mx + (1/4)\text{INC}, \tag{25}
$$

where the second inequality follows from $x \notin S(\sum_{i=1}^m [z_{\text{sdp}}]_i A_i, (1/4)\text{INC})$. Now observe that

$$
x^\top\Big(M + \sum_{i=1}^m [z_{\text{round}}]_i A_i\Big)x \geq x^\top Mx
$$

$$
> \lambda_{\min}\Big(M + \sum_{i=1}^m [z_{\text{sdp}}]_i A_i\Big) - (1/4)\text{INC}
$$

$$
= \lambda_{\min}(M) + (3/4)\text{INC}
$$

$$
\geq (3/4)(\lambda_{\min}(M) + \text{INC})
$$

$$
= (3/4)\lambda_{\min}\Big(M + \sum_{i=1}^m [z_{\text{sdp}}]_i A_i\Big)
$$

$$
\geq (3/4)\text{OPT} \geq (1/4)\text{OPT}
$$

where the strict inequality follows from rearranging (25), while the first and last equalities follow from the definition of INC.

**Case 2.2.** Suppose that $x \in S(\sum_{i=1}^{m}[z_{\mathrm{sdp}}]_i A_i, (1/4)\mathrm{INC})$. Then we have that

$$
x^\top\Big(M + \sum_{i=1}^{m}[z_{\mathrm{round}}]_i A_i\Big)x > \lambda_{\min}(M) + (1/4)\mathrm{INC}
$$

$$
\geq (1/4)(\lambda_{\min}(M) + \mathrm{INC})
$$

$$
= (1/4)\lambda_{\min}\Big(M + \sum_{i=1}^{m}[z_{\mathrm{sdp}}]_i A_i\Big)
$$

$$
\geq (1/4)\mathrm{OPT},
$$

where the strict inequality follows from $x^\top M x \geq \lambda_{\min}(M)$ (as $x \in S^{n-1}$) and (23), and the equality follows from the definition of INC. $\qquad\square$

