# OpenReview forum: "Simple Randomized Rounding for Max-Min Eigenvalue Augmentation"
_ICML.cc/2025/Conference — ICML 2025 poster_

### Official Review · Reviewer_Zjro · 2025-03-05

**Overall Recommendation:** 3

**Summary:**

This paper researches the $\textbf{max-min eigenvalue augmentation}$ problem: given symmetric PSD matrices $M, A_1, \cdots, A_m \in \mathbb{R}^{n \times n}$ and a positive integer $k < m$, the goal is to solve the following optimization problem $$\max_{z \in \\{0, 1\\}^m, \\|z\\|_0 \le k} \lambda\_{\min} \left( M + \sum\_{i=1}^{m} z_i A_i \right),$$which is a general case of $\textbf{Bayesian E-optimal design}$ and $\textbf{Maximum algebraic connectivity augmentation}$. Since this problem is NP-hard, this paper turns to solving its relaxed version $$\max\_{z \in [0, 1]^m, \eta \in \mathbb{R}} \left\\{\eta \bigg| M + \sum\_{i=1}^{m} z_i A_i \succeq \eta I, \sum\_{i=1}^{m} z_i \le k \right\\},$$which is an SDP problem, and utilizes the randomized rounding technique to get an approximate solution. Specifically, let $(z\_{\text{sdp}}, \eta\_{\text{sdp}})$ be the returned solution, $R := \max \\{ \text{tr}(A_i) \\}\_{i=1}^{m}$, and $$\text{INC} := \lambda\_{\min} \left(M + \sum\_{i=1}^{m} [z\_{\text{sdp}}]_i A_i \right) - \lambda\_{\min}(M).$$This paper establishes the guarantee that if $\text{INC} = \Omega( R \ln k)$, the randomized rounding method yields a constant-factor approximate solution to the original problem.

**Claims And Evidence:**

Yes, this paper provides a strict proof for the main theorem.

**Essential References Not Discussed:**

None.

**Experimental Designs Or Analyses:**

None.

**Methods And Evaluation Criteria:**

This paper is purely theoretical and doesn't include experiments.

**Other Comments Or Suggestions:**

(1) The definitions of quantities $R$ and $W$ only appear in Abstract and Introduction (Line 184). It would be helpful to repeat them when they appear in subsequent sections.
(2) In Line 92, for the maximum algebraic connectivity augmentation problem, it should be $M = \mathbf{1} \mathbf{1}^\top$ since $L$ is the Laplacian matrix corresponding to $A_e$.
(3) In Line 342, " It also of interest" -> "It is also of interest".

**Other Strengths And Weaknesses:**

$\textbf{Strengths:}$ (1) The core technical contribution is a novel intrinsic dimension concentration inequality for the minimum eigenvalue of a sum of random PSD matrices, filling a gap in matrix concentration inequalities.
(2) This work proves that the proposed method provides a constant-factor approximation when the optimal increase in the minimum eigenvalue is sufficiently large.
(3) Unlike many prior works, the approach accommodates augmentation matrices of arbitrary rank rather than just rank-one matrices.
$\textbf{Weaknesses:}$ (1) The theoretical guarantee holds only when the optimal increase is large enough, which might not always be the case in practice.
(2) There is no empirical evaluation or computational study to support the practical performance of the method.

**Questions For Authors:**

The max-min eigenvalue augmentation (MMEA) is closely related to the Bayesian E-optimal design (BED) and the maximum algebraic connectivity augmentation (MACA). Could you compare this paper with existing works on BED and MACA in terms of theoretical guarantees and running time?

**Relation To Broader Scientific Literature:**

The max-min eigenvalue augmentation (MMEA) problem studied in this paper can be taken as a generalization of the Bayesian E-optimal design (BED) and the maximum algebraic connectivity augmentation (MACA). However, there are two different aspects: (1) the augmentation matrix $A_i$, $i \in [m]$ is symmetric PSD, while it's a rank-one matrix in BED and MACA; (2) MMEA imposes the constraint $k < n$, which differs from the settings in BED and MACA.

**Theoretical Claims:**

The proofs seem correct since I didn't read all the proofs carefully.

---

> ### Author Rebuttal · Authors · 2025-04-01
>
> Thank you very much for taking the time to review our manuscript! Below we include our responses for each relevant section.
>
> $\text{Other Strengths And Weaknesses:\}$
> 1. Yes, the optimal increase might not always be sufficiently large in practice. Although, if it is not sufficiently large, then it might also be the case that it is not worth the cost of running design experiments (BED) or removing edges (MACA). Perhaps we should mention something like this in the paper, but we also do not want to give the impression of overselling our work.
> 2. We would be happy to add experiments (that we have already ran). Please see our response to reviewer gow2 (specifically, bullet number 4) for further discussion.
>
> $\text{Other Comments Or Suggestions:\}$
> 1. We agree and will repeat them in subsequent sections.
> 2. $L$ is the Laplacian matrix of the graph given in a problem instance (i.e., the graph without any edges removed). Taking $M = \mathbf{1}\mathbf{1}^{\top}$ corresponds to the case in which the given graph is empty.
> 3. We will update this typo.
>
> $\text{Questions For Authors:\}$
>
> In the bullets below, we discuss existing theoretical guarantees and run times for BED and MACA, respectively. Note that all work that we discuss considers the case in which the augmentation matrices are rank-one matrices.
> - BED. Work on BED focuses on the setting in which $k \geq n$. Most notably, (Allen-Zhu et al.. 2021) develop an algorithm that provides a $(1-\epsilon)$-approximation under the condition that $k = \Omega(\frac{n}{\epsilon^2})$. Because we focus on the setting in which $k < n$, our theoretical guarantee is not comparable. Regarding run times, their algorithm runs in $\tilde{O}(mn^2)$ time, while our algorithm runs in $\tilde{O}(m)$ time. That said, the main bottleneck is solving the SDP relaxation, so this runtime comparison is not relevant, unless one developed a method for approximately solving the relaxation in better than $\tilde{O}(mn^2)$ time.
> - MACA. (Kolla et al., 2010) provide guarantees for an algorithm that hold in both the $k < n$ and $k \geq n$ settings. Their algorithm provides a constant-factor approximation  under the assumption that the optimal increase is $\Omega(1)$, whereas our algorithm provides a constant-factor approximation under the assumption that $\Omega(\ln k)$, as $R = \sqrt{2}$ in this case. The runtime of their algorithm is again larger than simple randomized rounding’s runtime.
> We are happy to include this discussion in the manuscript. Additional references could be discussed, but these are the most relevant to your question.

---

### Official Review · Reviewer_gow2 · 2025-03-06

**Overall Recommendation:** 4

**Summary:**

In this work, the authors provided a new algorithm for the max-min eigenvalue augmentation problem. The method is able to achieve a constant approximation to the optimal value with a constant probability, given that the augmentation improvement is sufficiently large. The results are established by proving an extension of the matrix concentration inequality. Please see the following sections for my detailed comments.

**Claims And Evidence:**

Due to the time limit, I did not check the correctness of the theory, except those briefly mentioned in the main paper. The theoretical claims seem correct by checking the main paper.

**Essential References Not Discussed:**

N/A.

**Experimental Designs Or Analyses:**

N/A.

**Methods And Evaluation Criteria:**

The methods and evaluation criteria make sense to me.

**Other Comments Or Suggestions:**

- Line 275, left column: If $\beta$ is larger, I think only $L_1(\beta)$ will have a smaller dimension? This seems to be consistent with the fact that Lemma 3.2 only considers $L_1(\beta)$.

- It would be better if the authors could explain how the bound changes from the dimension $n$ in Theorem 2.5 to the intrinsic dimension. The same comment applies to Lemma 3.3.

- I wonder why the probability bound in Corollary 4.3 is 29/64 instead of 61/128?


- As pointed out by the authors, it would be important to include empirical results of the proposed algorithm. Although the contributions of this paper is on the theory side, it is helpful to show the value of the proposed algorithm by exhibiting the empirical performance. It would be particularly interesting to compare the new algorithm with existing algorithms designed for a specific application, such as the Bayesian E-optimal design problem and the maximum algebraic connectivity augmentation problem.

**Other Strengths And Weaknesses:**

Please see my comments in other sections.

**Questions For Authors:**

Please see my comments in other sections.

**Relation To Broader Scientific Literature:**

This paper is related to the topic of ICML conference and should be interesting to audiences from machine learning and numerical methods fields.

**Theoretical Claims:**

Due to the time limit, I did not check the correctness of the theory. The theoretical claims and the proofs in the main paper seem correct.

---

> ### Author Rebuttal · Authors · 2025-04-01
>
> Thank you very much for taking the time to review our manuscript! Below we include our responses for each relevant section.
>
> $\text{Other Comments or Suggestions:\}$
> 1. Yes, nice catch, we should have 1 instead of j here; we can make this update.
> 2. We agree that the exposition would benefit from this discussion. Regarding Lemma 3.2, the dimension of $L_1(\beta)$ is bounded by the intrinsic dimension. Regarding Lemma 3.3, there is a typo in the sentence preceding the lemma: it should read “Because Theorem 2.2. provides…” instead of “Because Theorem 2.4 provides…”. This sentence would then explain how intrinsic dimension comes into play.
> 3. Our reasoning is for $29/64$ is as follows. Taking $E_1$ and $E_2$ to be the events of interest, we have
> $$P(E_1 \cap E_2) = P(E_1) + P(E_2) - P(E_1 \cup E_2) \geq 1/2 + 61/64 - 1 = 29/64.$$
> 4. We agree with you (and ourselves for that matter!). We have already run some synthetic  (maximum algebraic connectivity and bayesian optimal design) experiments along these lines. We would be happy to include them, assuming that it is within scope; there is room for a small additional section, but we are not sure if it is standard to incorporate something like this at this point. We compare against Federov’s exchange algorithm because (1) the algorithm is the primary algorithm used in state-of-the-art software implementations, and (2) the algorithm can readily be applied in the general-rank and $k < n$ setting. Our main observations: Federov’s algorithm typically provides a better approximation, but it can take a significantly longer time to find an approximation that is on par with simple randomized rounding (even accounting for the SDP solve). Perhaps this would be better suited for future work.

---

> > ### Comment · Reviewer_gow2 · 2025-04-08
> >
> > I would like to thank the authors for the rebuttal! I would suggest the authors include the experiment results into the supplementary materials if there is not enough room in the main manuscript. I am happy to increase my rating.

---

### Official Review · Reviewer_XV8x · 2025-03-13

**Overall Recommendation:** 4

**Summary:**

This paper studies the max-min eigenvalue augmentation problem: choose a subset of at most $k$ PSD matrices $A_i$ to augment a PSD matrix $M$ to maximize the minimum eigenvalue of $M + \sum_i A_i$.

This problem generalizes the Bayesian E-optimal design (where certain experiments need to run together) and maximum algebraic connectivity augmentation problems (to $k < n$).

This paper gives a simple randomized rounding method of an SDP relaxation of the max-min eigenvalue augmentation problem, and proves that if the optimal increase (OPT - $\lambda_{min}(M)$) is sufficiently large, then the rounding method gives a constant-factor approximation.

The analysis depends on a new Chernoff-type matrix concentration inequality for the minimum eigenvalue of a sum of independent random PSD matrices.

**Claims And Evidence:**

In this theoretical papers, all claims are proved, or citations are provided.

**Essential References Not Discussed:**

I am not aware of essential references not discussed in this paper.

**Ethical Review Concerns:**

No ethical review needed.

**Experimental Designs Or Analyses:**

This paper does not have experiments.

**Methods And Evaluation Criteria:**

This is a theoretical paper with proofs but not evaluation.

**Other Comments Or Suggestions:**

This paper is very well written, and I don't have suggestions.

**Other Strengths And Weaknesses:**

(As stated in the paper and in the summary above,) the randomized rounding algorithm is very simple: include a matrix $A_i$ with probability exactly the value in the SDP relaxation. And the analysis appears to be intuitive.

However, this reader may appreciate some discussion on the technical restriction on the optimal increase ($OPT - \lambda_{min}(M) = \Omega(R\ln k)$): whether this restriction is necessary (e.g., due to lower bounds), or whether it arises naturally in practice.

**Questions For Authors:**

For the technical restriction on the optimal increase ($OPT - \lambda_{min}(M) = \Omega(R\ln k)$), besides “it works”, is it necessary (e.g., due to matching lower bounds), or is it a mild assumption in practice?

**Relation To Broader Scientific Literature:**

This paper proves a new Chernoff-type matrix concentration inequality for the minimum eigenvalue of a sum of independent random PSD matrices, while the literature only has such inequality for the maximum eigenvalue.

This paper gives an approximation algorithm for the max-min eigenvalue augmentation problem, which generalizes the Bayesian E-optimal design and maximum algebraic connectivity augmentation problems.

**Theoretical Claims:**

I checked the claims up to Section 2, and did not find any problems.

---

> ### Author Rebuttal · Authors · 2025-04-01
>
> Thank you very much for taking the time to review our manuscript! Below we include our responses for each relevant section.
>
> ${\bf \text{Questions For Authors:}}$
>
> We left the lower bound as a direction for future work, but since our original submission, we have realized that it is tight and would be happy to include this result. The result more-or-less follows from the fact that the matrix Chernoff inequality for the minimum eigenvalue (i.e., Theorem 2.4) is sharp (see 5.1.2. of Tropp). More specifically, the idea is to take $M$ to be a diagonal matrix whose first $n-k$ diagonal entries that are large and whose last $k$ other diagonal entries equal to $0$. We restrict the augmentation matrices to the las $k$ dimensions and take them as specified in Tropp. We could include the result as a short proposition and place its proof in the appendix.

---

### Official Review · Reviewer_4TVg · 2025-03-13

**Overall Recommendation:** 4

**Summary:**

This paper presents a matrix concentration on the minimum eigenvalue of the sum of PSD matrices, where the lower bound is parameterized via a generalization of the intrinsic dimension of the expected sum of the matrices. This complements the existing upper bounds e.g. from Joel Tropp's monograph on matrix concentration.

The paper shows that this concentration can be used to prove that a simple SDP relaxation + randomized rounding scheme achieves a constant factor approximation to the "max min eigenvalue augmentation problem" with constant probability in a regime where previous results did not give guarantees. This max min eigenvalue augmentation problem is a generalization of certain optimal design and resilient graph design problems.

While the regime that this paper studies appears to be less common invocations of these optimal design and graph problems, but seem like natural problems to study.

Paper is entirely theory; no experiments at all.

**Claims And Evidence:**

The paper is entirely theory.

The paper can be partitioned into two parts: proving the matrix concentration result, and proving that the concentration result is helpful for the max min eigenvalue augmentation problem.

Most of the paper focuses on the former, and shows how a certain partitioning of the unit sphere plus the existing matrix concentration literature can be used to prove this new minimum eigenvalue concentration result. It's a short and clever series of arguments.

There's a smaller section of the paper that argues how the SDP relaxation + randomized rounding algorithm provides a good (constant factor) solution to the max min eigenvalue augmentation problem, which seems to rely on relatively standard machinery as far as I can tell. It's good, but the emphasis is on the matrix concentration imo.

**Essential References Not Discussed:**

N/A

**Experimental Designs Or Analyses:**

No experiments.

**Methods And Evaluation Criteria:**

There's no empirical evidence in this paper.

Not a problem per se, just a statement.

**Other Comments Or Suggestions:**

List of typos and recommended edits. Feel free to ignore anything and everything here.

1. [Line 87, left] Shouldn't L be a PSD matrix? So shouldn't you have to subtract the all-ones matrix?
2. [Lines 95-101, right] While true, I'm not clear what's the point of the end of this paragraph.
3. [Lines 126-132, left] This "larger/smaller" language is awkward throughout. Say something like "If INC increases, then the number of quadratic forms x'Ax that are Omega(INC) decreases. More precisely, letting S = {...} be the set of directions (we refer to unit vectors as directions), we have that for any gamma ...."
4. [Lines 160-162, left] This break between paragraphs is awkwards and repetitive. Rephrase.
5. [Figure 1] Add more to the caption. Explain that $\alpha = \gamma INC$ maybe, and that increasing $\gamma$ makes the black boundary smaller. Also, make the colors print better in black and white
6. [Line 131, right] Define the intrinsic dimension you use directly with a clear definition. intdim(A,alpha) = tr(A)/alpha.
7. [Line 202, left] Specify page 61 for the thm statement, since it's not exactly the statement of thm 5.1.1. Same for theorem 2.4.
8. [Theorems 2.1, 2.2, 2.4, 2.5] I'd try to be more consistent between always writing these bounds in terms of t or 1-eps. Either is fine, but being more consistent would help.
9. [Line 210, right] This leftrightarrow symbol is kinda confusing here, because it sounds like you're more strongly asserting this iff claim. I would remove the leftrightarrow and replace it with parenthesis, so it instead says something like "(i.e. X > (1-eps) E[X])".
10. [Line 229, left] Remove "a"
11. [Line 230, left] Remove first "i \in [m]"
12. [Thm 3.1] Discuss when such an alpha exists. Because it doesn't always exist.
13. [Line 271, left] remove subscript on X_i in E[X_i].
14. [Line 291, left] Subscript on L should be 1 not j.
15. Consider having a more concrete theorem statement for E-optimal design or algebraic connectivity augmentation problem. It's just a corollary. But it'd be good to write out.

**Other Strengths And Weaknesses:**

The paper has nice writing, especially in its proof sketches. It's pretty intuitive, and feels nicely self-contained as well.

**Questions For Authors:**

See theory discussion -- what's the right way to think of the underlying concentration argument?

**Relation To Broader Scientific Literature:**

This concentration result seems like it might be a powerful short little result. This is why I harped on it in my review above. I don't fully appreciate how this result compares to the existing machinery, and I think the paper would benefit from having more of a discussion on this point.

I'd like to better understand that concentration in order to better understand the potential broader applicability of such a result.

**Theoretical Claims:**

I checked some of the proofs in detail, but not all. They seem essentially correct.

My main interest in examining the theoretical claims is in better understanding how the core novel concentration result, Theorem 3.1, compares directly with the existing results on both 1) minimum eigenvalue concentration without an intrinsic dimension term and 2) maximum eigenvalue concentration with an intrinsic dimension term. The claim seems to be correct, but I don't have a very clear sense of how it really compares to the prior work.

In particular, there's a few nonstandard terms in here:
1. The definition of intrinsic dimension used in their result is a generalization of the standard definition. In particular, the intrinsic dimension of a matrix, as defined in the Tropp monograph, is the ratio of trace to spectral norm. In this paper, it's the ratio of the trace to some parameter $\alpha$ which is less than or equal to the spectral norm. It's not nearly as intuitive to think about what this generalized intrinsic dimension is or should be.
2. They don't directly tackle the minimum eigenvalue problem. Instead, they do something more general I suppose. They have a parameter $\alpha$. They show that for all vectors $\vec v$ such that $\vec v^\intercal E[X] \vec v \geq \alpha$, we have $\vec v^\intercal X \vec v \leq (1-\varepsilon)\alpha$ with high probability, where $X$ is our sum of independent PSD matrices. This concentration result only holds for values of $\alpha$ between $O(L / \varepsilon^2)$ and $O(\| E[X] \|_{op})$, where L is a uniform upper bound on the operator norms of each summand. Further, the failure probability scales with $\frac1{\varepsilon^2 \alpha}$. When $\alpha = \| E[X] \|_{op}$, this should recover the usual definition of intrinsic dimension, but in this case very few vectors $\vec v$ will be covered in this theorem statement. It's not made 100% clear how this result translates to a genuine minimum eigenvalue guarantee, and I think the paper would benefit from a bit more discussion of this.
3. I note that the authors do have some light discussion of this, but it feels insufficient to me. Adding an appendix to think about the shape of this concentration result for (e.g.) and IID sum of terms and meditating on when a minimum eigenvalue guarantee follows would be rather beneficial in my perspective.

But the claims seem to be correct. The core results just needs more discussion imo.

---

> ### Author Rebuttal · Authors · 2025-04-01
>
> Thank you very much for taking the time to review our manuscript! Below we include our responses for each relevant section. We focus most of our response on your main line of inquiry.
>
> ${\bf \text{Theoretical Claims/Other Strengths and Weaknesses:}}$
>
> These are greats points. In what follows, we aim to shed light on them and discuss how we can address them in the manuscript. Hopefully this ties everything together.
>
> Consider [Lines 144-156, right]. These lines state that the minimum eigenvalue is a special case of the main quantity of interest when $\alpha = \lambda_{\mathrm{min}}(\mathbb{E}[X])$, as you point out in your 2nd comment under Theoretical Claims. While this is the case, our results do not imply anything interesting when $\alpha = \lambda_{\mathrm{min}}(\mathbb{E}[X])$; this seems to be what you are asking about in your 2nd and 3rd comment. Our writing then seems to be unintentionally misguiding. Our concentration inequality is only interesting when $\alpha > \lambda_{\mathrm{min}}(\mathbb{E}[X])$. Per this work, it is best to think of main random quantity $\min_{x \in S(\mathbb{E}[X],\alpha)}x^{\top}Xx$ as being intimately related to the minimum eigenvalue $\lambda_{\mathrm{min}}(X)$, but not ever the minimum eigenvalue itself. We can add clarification here as well as a forward references to later parts of the work for more detailed clarification (specifically, see the paragraph that follows the next two.)
>
> Regarding your comment about intrinsic dimension, we can modify the discussion of Remark 2.3. Our notion of intrinsic dimension provides an upper bound on Tropp’s intrinsic dimension. The upper bound is convenient for lower bounding $x^{\top}Xx$ over $x \in S(\mathbb{E}[X],\alpha)$, rather than upper bounding $x^{\top}Xx$ over $x \in S^{n-1}$, but it also generalizes Tropp’s definition, so we refer to it as intrinsic dimension.
>
> As you point out, it would be nice to understand how the novel concentration result compares with the minimum (and maximum) eigenvalue concentration result. In this spirit, it is natural to wonder why we cannot just apply the existing minimum eigenvalue concentration result? To position the reader’s thinking, we can explain after Theorem 2.4 why we cannot upper bound (6) with the matrix Chernoff inequality presented in Theorem 2.4. If there is a direction $x$ per which $x^{\top}\mathbb{E}[X]x = 0$, and hence $\lambda_{\mathrm{min}}(\mathbb{E}[X]) = 0$, then Theorem 2.4 provides us with a trivial guarantee. The existence of such a direction, however, should not impact upper bounding (6). One might hope that we can directly apply Theorem 2.4 to the set $S(E[X],\alpha)$, but the set is not a subspace (as seen in Figure 1), so we cannot do this. Although, this idea is exploited in the proof of Theorem 3.1; see Lemma 3.2.
>
> To address your point that the core result needs more discussion (and positioning with respect to the other concentration results), we can include the following discussion following Theorem 3.1.
> - Theorem 3.1 tells us that we can guarantee that $x^{\top}Xx$ is larger on a smaller set of directions $x \in S(\mathbb{E}[X],\alpha)$ per which the quadratic form $x^{\top}\mathbb{E}[X]x$ is larger. Accordingly, there is a tradeoff: For large values of $\alpha$, we can guarantee $x^{\top}Xx$ is larger, albeit in a smaller number of directions $x$. Ultimately, in the proof of Theorem 4.1, we choose $\alpha$ to optimize this tradeoff.
> - Consider the extreme case in which $\alpha = \lambda_{\mathrm{min}}(\mathbb{E}[X])$, which is still possible under the assumptions of Theorem 3.1. Using the fact that $\mathrm{tr}(\mathbb{E}[X]) \geq n \lambda_{\mathrm{min}}(\mathbb{E}[X])$, it then follows that Theorem 3.1 provides a weaker guarantee than Theorem 2.4.
>
> Following Theorem 4.1, we could discuss in slightly more detail how the proof follows from carefully choosing alpha based on the tradeoff inherent in Theorem 3.1 discussed above, i.e. taking $\alpha = \Omega(R ln k)$.
>
> ${\bf \text{Other Comments Or Suggestions:}}$
>
> We would like to include essentially all of your recommendations. We just make note of two spots in which we would like to keep the manuscript as is.
>
> - 1. Adding the matrix of all ones “pushes” the eigenvalue for the eigenvector of all ones to the top of the spectrum so that we can consider the minimum eigenvalue instead of the second smallest eigenvalue.
> - 9. At this point, we are hesitant to make this update so that we do not introduce any errors in the proofs that follow in later sections.

---

### Decision · Program_Chairs · 2025-05-01

**Decision:**

Accept (poster)

**Comment:**

This paper gives an approximation algorithm for a variant of the E-Optimal Design problem, called the Max-Min Eigenvalue Augmentation problem. The main result is a constant factor approximation via SDP relaxation and simple rounding when the augmentation is large enough. The main technical contribution is a new variant of the matrix Chernoff bound. The reviewers found the paper's result and techniques interesting, while noting some limitations (no experimental results, and primarily the condition on the augmentation being large). The paper would make a nice addition to the ICML program.